# Current Insights in the Mechanisms of Cobra Venom Cytotoxins and Their Complexes in Inducing Toxicity: Implications in Antivenom Therapy

**DOI:** 10.3390/toxins14120839

**Published:** 2022-12-01

**Authors:** Bhargab Kalita, Yuri N. Utkin, Ashis K. Mukherjee

**Affiliations:** 1Amrita School of Nanosciences and Molecular Medicine, Amrita Vishwa Vidyapeetham, Kochi 682041, India; 2Shemyakin-Ovchinnikov Institute of Bioorganic Chemistry, Russian Academy of Sciences, Moscow 117997, Russia; 3Institute of Advanced Study in Science and Technology, Guwahati 781035, India; 4Department of Molecular Biology and Biotechnology, Tezpur University, Tezpur 784028, India

**Keywords:** membrane perturbations, necrosis, cobra venom, antivenom neutralization, protein complex

## Abstract

Cytotoxins (CTXs), an essential class of the non-enzymatic three-finger toxin family, are ubiquitously present in cobra venoms. These low-molecular-mass toxins, contributing to about 40 to 60% of the cobra venom proteome, play a significant role in cobra venom-induced toxicity, more prominently in dermonecrosis. Structurally, CTXs contain the conserved three-finger hydrophobic loops; however, they also exhibit a certain degree of structural diversity that dictates their biological activities. In their mechanism, CTXs mediate toxicity by affecting cell membrane structures and membrane-bound proteins and activating apoptotic and necrotic cell death pathways. Notably, some CTXs are also responsible for depolarizing neurons and heart muscle membranes, thereby contributing to the cardiac failure frequently observed in cobra-envenomed victims. Consequently, they are also known as cardiotoxins (CdTx). Studies have shown that cobra venom CTXs form cognate complexes with other components that potentiate the toxic effects of the venom’s individual component. This review focuses on the pharmacological mechanism of cobra venom CTXs and their complexes, highlighting their significance in cobra venom-induced pathophysiology and toxicity. Furthermore, the potency of commercial antivenoms in reversing the adverse effects of cobra venom CTXs and their complexes in envenomed victims has also been discussed.

## 1. Introduction: Epidemiology of Cobra Bites in the World

Snake envenoming is a devastating health issue that affects millions of individuals across the globe, particularly in the developing countries of tropical and subtropical regions. Global epidemiological data suggest 81,000–138,000 snakebite-associated deaths from 1.8–2.7 million cases of envenoming yearly, thereby highlighting the urgency of research into this neglected public health concern [1,2]. Moreover, region-wise distribution of snakebite data suggests most of the cases of snake envenoming are concentrated in the rural areas of South Asia, Southeast Asia, and East Sub-Saharan African countries [3].

The majority of the medically critical venomous species of snakes belong to the Viperidae (340 species), Elapidae (360 species), and Atractaspidae (69 species) families [4]. Accordingly, the World Health Organization (WHO) has recognized venomous snakes under categories 1 and 2 depending upon their distribution, venom lethality, and incidences of envenoming and deaths. Category 1 medically essential species are of the highest medical importance, and the WHO has recognized at least 109 species under this category. Furthermore, it has been observed that most medically relevant species are represented by only a few genera. For instance, about 25 Category 1 medically important snake species in Africa belong to only five genera—*Naja*, *Dendroaspis*, *Echis*, *Bitis*, and *Cerastes* [5].

Cobras, represented by the genus *Naja* (nāgá, meaning ‘snake’ in Sanskrit), belong to the Elapidae family of snakes, and most of their species are classified under Category 1 by the WHO. There are nearly 30 species of cobra (*Naja*) (Table 1) that are widely distributed in Africa and Asia [6,7], and they are responsible for a considerable number of snake envenoming cases on these continents. For various reasons, most global epidemiological studies have not discretely recorded the exact estimates of cobra envenoming in Africa and Asia. While neurotoxicity is one of the vital clinical manifestations to distinguish cobra envenomation, other elapid snakes, for example, mambas (*Dendroaspis* sp.), kraits (*Bungarus* sp.), or even some viperids, for example, berg adders (*Bitis atropos*) [8], and Russell’s vipers (*Daboia russelii*) from southern India and Sri Lanka [9,10,11] have also been reported to inflict neurological disorders in patients. Therefore, identifying the inflicting species based solely on clinical manifestations might not always be accurate.

Moreover, except for Australia, no other countries currently employ snake venom detection kits to identify the encountered snake species [12]. Therefore, keeping track of exact numbers of envenomation by a particular snake species becomes quite challenging to identify the inflicting species taxonomically. In addition, discrepancies in epidemiological data on snake envenomation, either due to the absence of a cohesive and systematic analysis of snakebite cases or because many envenomation cases, are not reported to hospitals, may not be ruled out [13]. Nevertheless, given the wide distribution of cobras and their lethal venoms, it is reasonable to believe that these elapids contribute significantly to the morbidity and mortality associated with snake envenomation in Africa and Asia. Furthermore, evidence from a few isolated studies from some Asian and African countries indicates that the frequency of cobra (genus *Naja*) envenomation largely varies between 8–30% of the total envenoming cases [14,15,16,17,18].

Compositionally, cobra venoms are predominated by low-molecular-mass (<20 kDa) enzymatic and non-enzymatic toxins (see Section 2 for details). In addition, among the non-enzymatic class, the three-finger toxin (3FTX) is the prominent protein (toxin) family that significantly contributes to cobra venom-induced pathophysiology and toxicity [10,19,20,21,22,23,24,25,26]. Notably, among the various sub-classes of 3FTXs, the proportion of cytotoxins (CTX) and post-synaptic neurotoxins (NTX) dictates the lethality of cobra venoms [24,27].

In this review, we focus on the diverse pharmacological mechanisms of cobra venom CTXs and their cognate complexes that contribute to cobra venom-induced lethality in victims. In addition, being low molecular mass toxins, the recognition and neutralization of CTXs are critical for efficient in-patient treatment of cobra-envenomed patients. Therefore, we highlight commercial antivenoms’ efficacy in immuno-recognition, thereby neutralizing cobra venom CTXs and their cognate complexes. We searched the relevant literature on cobra venom cytotoxins from the period of 1960 to date using public search engines such as ScienceDirect (https://www.sciencedirect.com/, accessed on 26 October 2022), Google Scholar (https://scholar.google.com/, accessed on 26 October 2022), and PubMed (pubmed.ncbi.nlm.nih.gov, accessed on 26 October 2022). The keywords used for the literature search included cytotoxins, cardiotoxins, cytotoxin complex, cobra venom-induced necrosis, three-finger toxins, cytotoxicity neutralization, and necrosis neutralization.

## 2. Composition of Cobra Venom: A Summary

Biochemical and pharmacological analyses of crude *Naja* venom and their purified toxins are crucial in highlighting the variation in venom composition owing to the different geographical locations of these species [28,29]. However, these conventional methods are not apt for the characterization of non-enzymatic proteins, and with cobra venoms being primarily dominated by non-enzymatic toxins, a comprehensive cobra venom composition has yet to be unraveled. Recent developments in mass spectrometry-based snake venom toxin profiling have enabled thorough analysis of cobra venom composition. Proteomic findings have suggested that cobra venoms are predominated by enzymatic phospholipase A_2_ (PLA_2_) (~13–15 kDa) and non-enzymatic 3FTXs (~6–9 kDa); together, they constitute about 90% of the total cobra venom. Other enzymatic proteins in cobra venoms include L-amino acid oxidase, serine proteases, metalloproteases, phosphodiesterases, 5′-nucleotidases, cholinesterases, phospholipase B, hyaluronidases, and aminopeptidases. In addition, the minor non-enzymatic classes of toxins in cobra venoms include Cobra venom factors, Kunitz-type serine protease inhibitors, nerve growth factors, cystatin, natriuretic peptides, cysteine-rich secretory proteins, ohanin-like proteins or vespryns, vascular endothelial growth factors, and C-type lectins or snaclecs (reviewed by [4,30,31]).

Notably, the proportion of cobra venom CTXs was found to vary dramatically across different *Naja* species; it was ~13% in Taiwanese *N. kaouthia* venom, while it constitutes ~73% of *N. nigricollis* venom (Figure 1). In general, venoms from African spitting cobras have a higher proportion of CTXs than the Asiatic cobra ones, indicating geographical variation in snake venom composition. Interestingly, while the abundance of neurotoxins corresponds well to the severity of neurotoxicity in envenomed patients, the extent of local tissue-damaging effects of cobra venom do not correlate well with the proportions of CTXs, thereby suggesting variable cytotoxicity of the toxin isoforms as well as a contribution by other cobra venom toxins that can inflict local effects on their own or in complex with CTXs [32].

## 3. Discovery, Occurrence, and Classification of Cobra Venom CTXs

Intensive biochemical studies of venoms began in the mid-sixties of the 20th century. Their research methods were reductionist, i.e., to divide a venom into separate components and to study each one separately. This approach made it possible to detect the most abundant venom toxins. Thus, since cytotoxins are among the most represented toxins in cobra venoms, they were among the first to be detected and isolated. These toxins do not have enzymatic activity and are strongly basic polypeptides. The first cytotoxins were isolated from venoms in the late 1960s. Since these toxins exhibit an array of biological activities, they have therefore received different names reflecting the observed activity. They have been designated as cardiotoxin, skeletal muscle depolarizing factor, cobramine A and B, cytotoxin, toxin γ, direct lytic factor (DLF), peak 12 B, and others [48].

It should be noted that the cardiotoxic activity of cobra venom, later assigned to CTX, was first reported in 1948 [49]. The DLF, a CTX, was isolated and purified from the venom of Ringhals (*Haemachatus haemachatus*) [50]. The cobramines A and B have been purified from the venom of the Indian cobra (*N. naja*) [51]. In 1969, cardiotoxicity was demonstrated for DLF, and the name cardiotoxin (CdTx) was introduced [52]. These toxins are now called CTXs or CdTxs; the former is used more often.

The CTXs are small proteins comprising 60–62 amino acid residues with four intramolecular disulfide bridges [53]. Conventionally, they belong to the 3FTX family. Structurally, these molecules represent a globular core stabilized by disulfide bonds from which three distinct polypeptide loops or “fingers” emerge (Figure 2). It is to be noted that a slight variation in amino acid sequences in CTXs from different cobra species can result in differential cytotoxicity [54]. Moreover, the Asp57 to Asn57 (D57N) mutation has been demonstrated to perturb the structure of CTX molecules at a neutral pH [55]. More detailed information about CTX structure can be found in the review by Konshina et al. [56].

More than one hundred CTX sequences have been deposited in the UniProt protein database (https://www.uniprot.org/uniprotkb?query=cytotoxin%20AND%20(taxonomy_id:8570)) (accessed on 29 September 2022). They were identified mainly in the venom of cobras from the *Naja* genus. A few CTXs were also isolated from the venom of Ringhals or ring-necked spitting cobra from the *Hemachatus* genus [50,58,59]. Two CTX homologs were isolated from the venom of the common shield cobra *Aspidelaps scutatus fulafulus* [60], and three CTX-like proteins were isolated from the venom of a king cobra (*Ophiophagus hannah*) [61]. Interestingly, Thakur et al. [62] reported the isolation of a venom complex containing a 27.6 kDa protease and a CTX-like component named Rusvitoxin (6.6 kDa) from Indian Russell’s viper (*Daboia russelii*) venom that shared significant sequence similarity with CTXs from *Naja* sp.

CTXs form a reasonably homogeneous group of toxins. Within this group, CTXs are classified into two types, namely S-type and P-type. This classification is based on the position of two amino acid residues in toxin amino acid sequences. All S-type CTXs contain a serine residue at position 28, while in all P-type CTXs, a proline residue stands at position 30 [63] (Figure 3).

Another classification of CTXs was suggested based on phylogenetic analysis [64]. This classification divided all CTXs into the Type IA and Type IB groups. The Type IA group includes all known CTXs except three CTXs from the Ringhals venom. The Ringhals CTXs constitute the Type IB group. While the classification of CTXs into S-type and P-type is supported by the data about their differences in biological activity [65,66,67], no similar data are available for Type IA and IB groups.

## 4. Complex Formation of Cobra Venom CTXs with Other Components of Venoms

Toxin synergism is an intriguing phenomenon observed in snake venom toxins, yet it is relatively unexplored. Condrea et al. [68] demonstrated the first evidence of cobra venom’s synergistic effects. They explained that although individual chromatographic fractions of PLA_2_ and CTX of *N. naja* venom are devoid of the hemolytic activity or are weakly hemolytic, a combination of the two fractions induced profound hemolysis. In 1995, Chaim-Matyas et al. demonstrated synergistic hemolysis by a sub-lytic concentration of CTX P4 from the venom of *N. nigricollis* combined with non-lytic concentrations of PLA_2_s isolated from the venoms of *N. nigricollis*, *N. melanoleuca*, *N. atra*, *Bitis arietans*, *Pseudocerastes persicus*, and *Walterinnesia aegyptia* [69].

A similar observation on toxin synergism was reported for *N. kaouthia* venom PLA_2_ (NK-PLA_2_) and weak neurotoxins (kaouthiotoxins, KTXs). While both the purified toxins did not induce hemolytic activity on mammalian erythrocytes, the hemolytic potency of the KTX:NK-PLA_2_ complex was quite significant [70,71]. However, the exact stoichiometry of the interaction between KTX and NK-PLA_2_ was not determined in the study. Another hetero-trimeric cognate complex consisting of PLA_2_, NTX, CTX (1:2:1), and a trace amount of nerve growth factor was characterized in the venom of the Indian cobra *N. naja*. The complex exhibited marked synergism in cytotoxicity towards rat L6 myogenic cells, which has a profound pathophysiological significance in cobra venom-induced local tissue necrosis [72]. Another recent study hypothesized that toxin synergism between PLA_2_ and CTX isolated from the venoms of *N. nigricollis*, *N. melanoleuca*, and *N. mossambica* occurs via the formation of toxin hetero-oligomers that are better poised to interact with and perturb cell membranes [73]. These findings highlight that other venom toxins can interact with cobra venom CTXs, leading to overall augmentation in its cytotoxic effects. Although the exact stoichiometry of such complexes and the distinct mechanisms of action are still understudied, further research into this exciting field can aid in a better understanding of toxin synergism and its implication in cobra venom-induced pathophysiology and toxicity in the victim.

## 5. Pharmacological Mechanism(s) of Cobra Venom CTXs Vis-à-Vis Their Complexes: Correlation to Cobra Venom-Induced Pathophysiology and Toxicity

It is well established that CTXs interact and penetrate the cell membrane; the tips of all three loops are immersed at different degrees [56,74]. However, the exact molecular mechanisms of this interaction and further steps are not yet entirely understood. Moreover, apart from altering membrane characteristics, CTXs are also reported to induce cell cycle arrest, apoptosis, necrosis, and in a few cases, necroptosis (Figure 4; Table 2). The following sections dwell on the diverse mechanisms of action of CTXs that have been thus far established.

### 5.1. Cytolytic Action by Disruption of Membrane Integrity

The first step in the CTX–cell interaction is recognizing phospholipids’ micro-domains or rafts on the cell surface, for which membrane models are often used to identify this step. Earlier works about CTX–membrane interactions are discussed in the reviews [54,56,74], where several membrane-binding models of CTXs were suggested. The mechanism initiates with a non-insertion mode, in which electrostatic binding is predominant, followed by an edgewise penetration (insertion) mode with the incorporation of the tips of the loops into one of the membrane leaflets. This step is followed by the isotropic phase (toxin/lipid complexes), which characterizes membrane deterioration by the CTXs. Finally, the oligomeric state of CTXs was suggested with specific interaction between CTXs and the lipids’ head group and the possible formation of pores in the lipid bilayer [74]. Later studies added more details to the mechanism of CTX–membrane interactions, and they are outlined hereafter.

#### 5.1.1. Interaction with the Lipid Bilayer

Molecular dynamics simulations were used to map the CTX amino acid residues essential for the hemolytic activities of CTX1 from *N. naja* venom [75]. The authors constructed a heterogeneous pentameric erythrocyte membrane model to study the structural interactions between CTX1 and the lipid bilayer. Simulation results suggested that the involvement of two structural elements—‘head groove’ and ‘loop groove’ (of loop II)—are vital for establishing an interaction between CTX1 and the erythrocyte membranes.

Another simulation study employed coarse-grained and full-atom molecular dynamics analyses to predict the interaction of *N. oxiana* venom S-type CTX (CT1) with the phospholipid bilayers [76]. Notably, before simulation, the spatial model of CT1 was developed in dodecyl phosphocholine (DPC) micelles using high-resolution nuclear magnetic resonance (NMR) spectroscopy to mimic the structure of CT1 in a lipid environment. The molecular dynamics simulations were performed to understand the partitioning of the NMR-resolved CT1 structure in the DPC micelles and palmitoyl-oleoyl phosphatidylcholine (POPC) lipid membrane. Interestingly, CT1 exhibited distinct interactions with both lipid structures. In the case of DPC, CT1 inserted the tips of all three loops into the micelles, followed by a change in the frame at the end of loop II. Residue-level analysis suggested that the hydrogen bonding between Asp29 of CT1 with the tightly bound water molecule in loop II was lost. A new hydrogen bond has emerged, which involved Thr31 of CT1. However, in the presence of POPC membranes, the partitioning of CT1 proceeds primarily with either the tip of the loop I or both ends of loops I and II [76].

While the above studies investigated the structural interactions between CTXs and the lipid bilayer, an exciting report aimed to study the precise mechanisms of stepwise insertion of a P-type CTX (CT2) isolated from *N. oxiana* venom into the lipid membranes using a molecular dynamics simulation [80]. An atomistic level simulation predicted several discrete conformational changes in CT2 during insertion into lipid membranes. Furthermore, it was found that the hydrophobic regions of CT1 (formed by the tips of the three loops) are too bulky to penetrate the membrane all at once, and that loop I aids in the gradual immersion of the CTX molecule through the membrane. In addition, the study identified several lipid-binding sites in CT2, which facilitated the insertion of the toxin into the membrane [80].

More recently, the variability in the structures of the central loop (loop II) of S-type CTX 13 from *N. naja* venom (CT13Nn) between “water” and “membrane” conformations was explored using X-ray crystallography and molecular dynamics simulations [57]. Structural analyses suggested transforming from the “water” conformation to the “membrane” conformation in loop II of CT13Nn during its insertion into the lipid membranes. Notably, these conformational adaptations in loop II are not observed in P-type CTXs [57]. Findings from a few recent studies have also advanced our understanding of the interaction of CTXs with lipid membranes at an amino acid residue level, specifically, the importance of Asp residues of CTXs isolated from *N. atra* venom (CTX1 and CTX3). Semi-carbazide-modified Asp residues of CTX1 (Asp29 and Asp40) aggravated the ability of the CTX to cause permeability of lipid vesicles and cell membranes; however, native and semi-carbazide-modified CTX3 (Asp40) exhibited similar membrane-damaging activity. These findings suggest that when the negative charge at Asp29, located at the tip of loop II, is blocked, CTX1 adopts a structural conformation that exhibits profound interaction with cell membranes. On the contrary, Asp40, located distant from the tip of loop II, is not crucial for demonstrating membrane-perturbing activity but may be involved in the other pharmacological activities of the CTXs [103,104].

#### 5.1.2. Interaction with Some “Receptors” on The Cell Membrane

While conventional receptors specific to cobra venom CTXs have not been identified, a study by Lee et al. [85] described the presence of distinct heparin sulfate domains on the cell membranes of H9C2 rat cardiomyocytes and Chinese hamster ovary cells, which promotes endocytosis of two CTX isoforms (A2 and A4) isolated from the venom of the Taiwan cobra (*N. atra*). Notably, the isoforms interacted with heparin domains with varying degrees of sulfation for their internalization; while CTX A2 interacted with low-sulfated domains, CTX A4 preferred fully sulfated heparin domains for their internalization. Moreover, the internalization of CTX A2 and A4 was regulated by dynamin2, an essential GTPase protein required for caveolae- or clathrin-mediated endocytosis. Inhibiting this protein by a specific small molecule inhibitor, dynasore (C_18_H_14_N_2_O_4_), significantly reduced the internalization and cytotoxicity of the CTXs (Table 2) [85]. In addition, the cholesterol concentration of the cell membranes was also indicated to have an impact on the internalization mechanism; an elevated cholesterol concentration in the cell membranes augmented the endocytosis of both CTX A2 and A4 and vice-versa [85]. Furthermore, another study provided indirect evidence of the internalization of the Indian cobra *N. naja* venom CTX (CTX2a) in a complex with an acidic PLA_2_ and an NTX via specific binding of the PLA_2_ to the vimentin (an intermediate filament family of proteins) of rat L6 myogenic cells (Table 2) [72].

Interaction of CTXs with the cell membrane and/or specific domains results in alteration of the membrane’s permeability, conditionally resulting in membrane perforations, which perturbs ion transport and causes leakage of cell contents. Subsequent internalization of CTX molecules into the cell drives the destruction of lysosomes and mitochondria and the disruption of intracellular cascades.

#### 5.1.3. Destruction of Lysosomes and Mitochondria

After internalization of CTX molecules, they primarily localize in the lysosomes and mitochondria; however, the localization of internalized CTX complexes has not been studied in detail. Therefore, it remains to be elucidated as to whether or not the CTX complexes can also cross the cell membrane. Perturbation of lysosomal membranes leads to the release of cathepsins, thereby causing necrotic cell death. Several cobra venom CTXs have been demonstrated to act via such a mechanism (see Section 5.4 for details). Furthermore, when mitochondrial membrane integrity is challenged, the intrinsic apoptosis pathway of cell death is activated. Of note, CTXs bind specifically to cardiolipin to cause mitochondrial dysfunction, which can be structural and functional (see Section 5.3 for a detailed mechanism). While some CTXs activate apoptotic cell death by disrupting mitochondrial membrane permeability and promoting the release of cytochrome c into the cytosol, a few others (for example, S-type cardiotoxin VII4 from the venom of *N. mossambica*) have also been demonstrated to cause anomalous mitochondrial fragmentation, resulting in a reduction of oxidative phosphorylation and a decline in ATP production [105].

#### 5.1.4. Disturbance of Intracellular Cascades

Apart from destroying the lysosomes and mitochondria, some CTXs can interfere with the mitochondrial and other cellular signaling pathways, activating cell cycle arrest, intrinsic and extrinsic apoptosis, necrosis, and necroptosis. The mechanisms of dysregulated signaling are described in detail in later sections (Section 5.3 and Section 5.4). Interestingly, a lethal CTX from the venom of *N. kaouthia* (NK-CT1) has been predicted to interact with the oligonucleotide–human DNA topoisomerase II alpha complex using molecular docking. More specifically, the amino acid residues Met26, Val27, and Ser28 of NK-CT1 could interact with the nitrogenous bases of the oligonucleotide, which are in complex with DNA topoisomerase II alpha [97]. Nevertheless, the authors present no experimental evidence for these interactions.

### 5.2. Membrane Depolarization and Contraction

While disrupting membrane characteristics and altering intracellular signaling manifests as the pathological local effects of cobra envenomation, cobra venom CTXs also act on cardiomyocytes that contribute to cardiac arrest observed in cobra-envenomed victims. Therefore, they are also referred to as cardiotoxins (CdTxs). The cardiotoxic mechanisms have been thoroughly investigated for the two CTX isoforms from *N. oxiana* venom—S-type CTX1 and P-type CTX2. The force-frequency relationship (FFR) of myocardium contractility is an intrinsic regulatory mechanism of the heart that describes the frequency-dependent variations in cardiac contractile force without external stimuli [106]. During standard physiological heart rates, the FFR for most mammals is positive, while during a cardiac arrest, the FFR is found to be negative. Notably, the FFR under the influence of CTXs (or CdTxs) becomes entirely negative [77], which explains the onset of cardiac failure in cobra-envenomed patients.

The mechanism of the above phenomenon was investigated in a series of comparative analyses of CTX1 and CTX2 from *N. oxiana* venom. It was noted that at a concentration of 1 µg/mL, both toxins exhibit similar contractions of papillary muscles isolated from the right ventricles of rat hearts. However, when a higher concentration (2 µg/mL) was used, the P-type CTX2 subdued the muscle contraction entirely, whereas for the S-type CTX1, the contraction was only 50% of the control samples. Furthermore, in response to external stimulation (0.1 to 0.5 Hz frequency range), both the CTXs (1 µg/mL) induced an initial positive inotropic effect (an indication of the increased force of the heartbeat) in the rat papillary muscles. Nevertheless, at physiological stimulation frequencies (~1 to 3 Hz), the inotropic effect became entirely negative [77], which indicates a weakening of the heart’s contractions and a slow heart rate (bradycardia)—a clinical presentation observed in cobra-envenomed victims [107].

Recently, the cardiotoxic effects of CTX1 and CTX2 were also studied in an isolated rat heart, and the diverse parameters of heart contraction and function were recorded [66]. Results suggested that after a latency period of 3–8 min, both the CTXs (5 µg/mL) induce a marginal increase in the systolic pressure, which is followed by a prompt fall in the systolic pressure and a concomitant increase in the diastolic pressure until complete heart muscle contraction. While the slight increase in the systolic pressure was similar for both the toxins (up to 103 mmHg for CTX1 vs. 123 mmHg for CTX2), a significant difference in the increase of the diastolic pressure was recorded (104 mmHg for CTX1 vs. 181 mmHg for CTX2). Moreover, in the case of CTX2, the magnitude of cardiac contracture was significantly higher, and the cardiotoxic events developed faster (413 s vs. 676 s) than in CTX1, thereby suggesting that P-type CTX2 is a more potent CdTx compared to S-type CTX1.

The significance of calcium-dependent signaling for normal heart function has been appreciated for decades [108]. A few studies have unveiled the mechanisms involving changes in intracellular calcium ion (Ca^2+^) concentration that drives cardiac contractility in response to CTXs. Moreover, Ca^2+^ overload in the cardiomyocytes is one of the vital factors in the pathological effects of CdTxs [109]. However, CTXs employ diverse ways to elevate the intracellular Ca^2+^ concentration. A *N. atra* CTX was demonstrated to activate the L-type calcium channels for the influx of Ca^2+^ and activate the calcium-dependent signaling for the contraction of neonatal rat cardiomyocytes and adult rat aortic rings [86]. On the contrary, CTX1 and CTX2 from the venom of *N. oxiana* could induce contracture of papillary muscles isolated from the right ventricles of rat hearts even after blocking the L-type calcium channels with Nifedipine (2 μM), an L-type calcium-channel blocker. These data suggest that *N. oxiana* CTXs form non-selective pores in the cell membrane that facilitates the influx of Ca^2+^ and stimulation of the contracture instead of activating the L-type channels [67,77].

### 5.3. Activation of Cell Cycle Arrest and Apoptotic Cell Death Pathways

With recent advancements in CTX research, it is evident that apart from causing membrane perturbations, CTXs can promote cell cycle arrest and activate cell death signaling pathways to induce cytotoxicity. Cell cycle arrest is a regulatory process that halts the progression of a cell through the cell cycle during one of the typical phases (G1, S, G2, and M) to enable DNA damage repair before proceeding to cell division. However, irreparable DNA damages or other extra- and intracellular alterations promote activation of programmed cell death mechanisms (apoptosis) to avoid DNA damages in the cell or to eliminate unwanted cells. Notably, most of the mechanistic studies of cobra venom CTXs on cell cycle arrest and apoptosis are conducted in cancer cell lines to explore the anti-cancer properties of these toxins [54]. Nevertheless, these findings can be extrapolated to normal cells to describe the local effects of cobra envenomation, as CTXs often exhibit non-specific cytotoxicity.

CTX III (6.8 kDa) isolated from *N. atra* venom arrested the proliferation of HL-60 leukemia cells at the sub-G1 stage, as evident from flow cytometry analysis [88]. In addition, CTX III could also induce the arrest of K562 leukemia cells and Ca9–22, SAS, and CAL27 oral squamous carcinoma cell lines at the G2/M and S phases by downregulating the expression of cell cycle regulatory proteins, for example, cyclin B1, cyclin A, cell division cycle 25C (Cdc25C), and cyclin-dependent kinase 1 (Cdk 1) [89,90]. Similarly, a lethal CTX (6.7 kDa) isolated from Indian Monocled cobra (*N. kaouthia*) venom could halt the cell cycle progression in U937 (IC_50_ 3.5 µg/mL) and K562 (IC_50_ 1.1 µg/mL) leukemia cells at the sub-G1 stage (Table 2) [98]. In all of the above circumstances, cell cycle arrest eventually activated the extrinsic or intrinsic apoptotic pathways.

Apoptosis is a regulated cell death program driven by the activation of cysteine–aspartic proteases (caspases). Two distinct apoptotic mechanisms, the intrinsic and extrinsic pathways, allow the elimination of cells exhibiting abnormal physiology. While the extrinsic pathway is regulated by the activation of death receptors such as Fas, TNFR, and TRAIL, the intrinsic pathway is stimulated by stress-associated activation of pro-apoptotic proteins (Bax and Bak) that subsequently releases cytochrome c from the mitochondria to the cytoplasm for activation of the caspases [110,111]. The majority of the cobra venom CTXs characterized to date induce apoptotic cell death via the intrinsic pathway, which can be activated by the generation of reactive oxygen species (ROS), elevated calcium (Ca^2+^) influx into cells, modulation of pro- and anti-apoptotic proteins, and mitochondrial fragmentation [78,81,87,90,91,92,93,94,95,99,100].

Incubation of SK-N-SH human neuroblastoma cells with CTX3 (1 µM) and CTX4 (1 µM) from *N. atra* venom could induce ROS generation followed by alteration in mitochondrial permeability, cytochrome c release, and activation of caspases 3 and 9, that ultimately resulted in apoptotic cell death (Figure 4, Table 2) [87]. In addition, CTX3 (150 nM) was reported to induce Ca^2+^ influx in U937 human leukemia cells that resulted in the degradation of protein phosphatase 2A (PP2A) followed by subsequent phosphorylation of adenosine monophosphate-activated protein kinase (AMPK) [95]. Phosphorylated AMPK triggers mitochondrial fragmentation [112], releasing cytochrome c into the cytosol to activate the caspases. Moreover, several studies have shown that the elevated expression of pro-apoptotic proteins (Bad, Bax, endonuclease G) and concomitant downregulation of anti-apoptotic proteins (Mcl-1, Bcl-2, survivin, Bcl-XL and XIAP) in response to CTX3 (0.15—1 µM) from *N. atra* venom eventually resulted in activation of the intrinsic apoptotic pathway in Ca9–22 oral squamous cell carcinoma cells, MDA-MB-231 breast cancer cells, A549 lung cancer cells, colo 205 colorectal cancer cells, and K562 leukemia cells (Figure 4, Table 2) [90,91,92,93,94].

As has been discussed in Section 5.1, CTXs can interact with and penetrate through the cell membrane. After penetration into the cells, they interact with cardiolipins (a unique dimeric phospholipid found in mitochondrial walls) and become localized within the mitochondria. Notably, molecular dynamics simulation predicted the electrostatic association of Lys12 of the loop I and Lys30 of loop II of cardiotoxin VII4 from the venom of the Mozambique spitting cobra (*N. mossambica*) with phosphate groups of cardiolipin [105]. Similar simulation studies identified the critical amino acid residues of CTI and CTII from the venom of the Caspian cobra or the Central Asian cobra (*N. oxiana*) that interacts with cardiolipin and phosphatidylcholine of multilamellar liposomes that mimic mitochondrial membranes [113]. Subsequent infiltration of CTXs into the mitochondria perturbs the mitochondrial permeability and signaling, ultimately leading to mitochondrial fragmentation and stimulation of intrinsic apoptosis [114]. CTXs from *N. oxiana* (CTI and CTII), *N. sumatrana* (SumaCTX), and *N. haje* (NHV-Ic) were reported to activate intrinsic apoptosis in Hep-G2 hepatocellular carcinoma cells, HCT-116 colon carcinoma cells, HeLa cervical carcinoma cells, bovine cardiomyocytes, and MCF-7 breast cancer cells via this mechanism (Figure 4, Table 2) [78,81,99,100].

While most of the cobra venom CTXs are associated with the induction of intrinsic apoptosis, a CTX from *N. atra* (CTX1) was recently demonstrated to induce upregulation of expression of FasL and Fas, the proteins involved in the regulation of cell death, thereby activating the extrinsic apoptotic pathway in HL-60 and U937 leukemia cells. Mechanistic studies revealed activation of the Ca^2+^/NOX4/ROS/p38 MAPK signaling axis, thereby promoting c-Jun-mediated Fas and ATF-2-mediated FasL upregulation (Figure 4, Table 2) [82]. While apoptotic cell death induced by cobra venom CTXs has been thoroughly investigated, the same mechanism for CTX complexes remains elusive.

### 5.4. Activation of Necrotic and Necroptotic Cell Death Pathways

Necrosis is an inappropriate cell death and disintegration process resulting primarily from extreme stress, which can be either due to mechanical or intra- and extracellular biochemical alterations [115]. Necrotic cell death is often accompanied by cell membrane disruption leading to the release of cytoplasmic contents and inflammation [116]. On the contrary, necroptosis is a form of programmed necrosis regulated by activation of death receptors of the tumor necrosis factor (TNF) superfamily (for example, TNFR1, interferon receptors (IFNR), and the toll-like receptors (TLR3 and TLR4)) [117,118]. In addition, in some cases of necroptosis, the involvement of receptor-interacting protein (RIP) kinase has also been implicated [119,120]. However, necrosis and necroptosis are pathologically challenging to distinguish in envenomed patients.

CTXs representing nearly half of the cobra venom proteome in most of the *Naja* species significantly contribute to the activation and progression of necrosis and/or necroptosis at the proximities of the envenomed sites, which is one of the major clinical presentations of cobra envenomation. Internalization of CTXs and their complexes, followed by perturbation of lysosomal membrane integrity, seems to be one of the most prominent mechanisms of CTX-induced necrosis (Figure 4, Table 2) [113,121]. Disruption of the lysosomes results in the release of the lysosomal cysteine protease, cathepsin, which drives the cells into necrosis. Notably, a moderate amount of cathepsin B can also promote apoptotic cell death by stimulating the release of cytochrome c from the mitochondria; however, higher levels of cathepsin B are found to be exclusively associated with necrosis [81,122]. Moreover, the amount of cathepsin released from the lysosomes invariably depends upon the extent of exposure of cells to CTXs. Several CTX isoforms isolated from the venoms of *N. oxiana* and *N. atra* have been demonstrated to induce extensive damage to the lysosomal membranes (IC_50_ ranging from 4 to 55 µg/mL). Loss of lysosomal membrane integrity was accompanied by the release of large amounts of cathepsin B that subsequently resulted in necrosis of non-cancerous (16HBE human bronchial epithelial cells and MDCK Madin–Darby canine kidney cells) as well as cancerous cells (MCF-7 breast cancer cells; K562, HL-60, and P388 leukemia cells; H22 and HepG2 liver cancer cells; and DU-145 prostate cancer cells) [79,83].

Furthermore, the manifestation of dermonecrosis under in vivo conditions has also been established for purified CTXs from *N. atra* venom [96]. In addition, RP–HPLC fractions of the black-necked spitting cobra (*N. nigricollis*) venom containing CTX and PLA_2_ have also been shown to induce necrosis in CD-1 mice; however, any possible complex formation between these toxins was not investigated in detail [102].

Necroptosis, a pathological event similar to necrosis, has been reported only in a few cases of cobra venom CTXs. SumaCTX, a CTX isolated from the venom of *N. sumatrana* at a dose of 29.8 μg/mL, could activate necroptotic cell death in MCF-7 breast cancer cells via upregulation of peptidyl–prolyl isomerase (PPIase) and heat shock proteins (HSP90AB1 and HSP90AA1). However, the exact mechanism of the involvement of the TNFR, IFNR, or TLR for the induction of necroptosis was not explored (Figure 4, Table 2) [101]. Similarly, CTX1 isolated from *N. atra* venom was demonstrated to promote necroptosis in KG1a (IC_50_ 3.3 µg/mL) and HL-60 (IC_50_ 10.1 µg/mL) leukemia cells. Although the exact molecular mechanism of CTX1-induced necroptosis has not been investigated, treatment with necrostatin-1 however, a small molecule necroptosis inhibitor, was found to reverse the effects of CTX1 in the leukemia cells. In contrast, the caspase inhibitor Z-VAD-fmk did not affect the CTX1-induced cell death, thus indicating the necroptosis induction by CTX1 [84].

Although several studies have investigated the necrosis/necroptosis induced by cobra venom CTXs under in vitro or in vivo conditions, our understanding of the same processes for CTX complexes is nevertheless still limited. A classic example of synergistic cytotoxicity exhibited by cobra venom toxins is the kaouthiotoxin (KTX): NK-PLA_2_ complex isolated from *N. kaouthia* venom. While individual proteins of the complex failed to induce substantial hemolysis of human erythrocytes, the hemolytic potency of the complex was quite significant [70]. Another cognate complex isolated from the venom of *N. naja* containing PLA_2_, NTX, CTX (1:2:1), and a trace amount of nerve growth factor exhibited marked synergism resulting in augmented cytotoxicity towards rat L6 myogenic cells [72]. In this study, it was advocated that the 3FTXs of the complex first destabilize the phospholipid membranes of the myoblasts, thereby facilitating enhanced binding of the PLA_2_ on the phospholipid bilayers for extended membrane damage, ultimately resulting in cell death.

Interestingly, the complex exhibited preferential cytotoxicity towards different cell types; the highest toxicity was demonstrated against myoblast, followed by platelets and rat pheochromocytoma PC-12 neuronal cells [72]. The authors observed an enhanced release of creatine kinase (CK) and lactate dehydrogenase (LDH) from the rat myoblasts upon treatment of the complex, indicating cell lysis. Nonetheless, the exact mechanism for cytotoxicity, whether apoptotic or necrotic, was not proposed. Nevertheless, such a cognate complex in *N. naja* could explain the dermonecrosis observed in envenomed patients. Characterizing such complexes from other *Naja* species in the future will aid in a better understanding of the pathophysiology of dermonecrosis in cobra-envenomed patients.

## 6. A Comparative Study on Neutralization of Cobra Venom CTXs and Their Complexes with Commercial Antivenoms

Equine or other mammalian-derived antivenoms, either in a polyvalent or a monovalent form, remain the mainstay for treating snake envenomation. However, it is now well-appreciated by toxinologists around the globe that the currently available commercial antivenoms have certain limitations [123]. Among others, the sub-optimal recognition and neutralization of low molecular mass (<20 kDa) venom toxins has been a long-standing hurdle against effective in-patient management of snake-envenomed patients. Due to their smaller size, these low molecular mass toxins cannot elicit a robust immune response in immunized horses/other animals [11,22,23,24,124,125,126,127,128,129]. As a result, the antivenoms thus produced do not contain sufficient antibodies to immuno-recognize and subsequently neutralize these toxins. In addition, variation in cobra venom proteome composition owing to a geographical location is another crucial factor that significantly contributes to poor antivenom performance in certain regions [22,23,24,41,130,131].

Cobra envenomation primarily manifests two distinct pathologies, neuromuscular paralysis and local tissue necrosis. In addition, as discussed earlier, CTX is the prominent protein family in *Naja* venoms that contributes to local necrotic effects. Although the local tissue damage may not always be life-threatening, it frequently results in irreversible morbidity in the victims. Once local tissue necrosis initiates, the patients must undergo fasciotomy, debridement, and in severe cases, amputation of the affected extremities. These surgical consequences severely affect the patient’s life even after recovery. Therefore, immuno-recognition and neutralization of low molecular mass cobra venom CTXs (~6 kDa) is a critical concern for the efficient hospital management of cobra-envenomed patients, especially in countries where cobra envenomation is a constant threat [22,23,24]. Furthermore, the circumstances can aggravate while dealing with envenomation cases by those *Naja* species in which CTXs contribute a significant proportion of the venom proteome, such as *N. nigricollis* (72.8%), *N. katiensis* (62.7%), *N. pallida* (64.9%), *N. mossambica* (67.7%), etc. [25].

The immuno-recognition and neutralization of cobra venom CTXs by commercial antivenoms has been investigated using diverse approaches, including western blot, enzyme-linked immunosorbent assay (ELISA), antivenomics, and under in vitro conditions using cell line models as well as in vivo animal models. Petras and his co-workers [25] investigated the toxin composition of the African spitting cobras (*N. nigricollis*, *N. katiensis*, *N. nubiae*, *N. mossambica*, and *N. pallida*) using LC-MS/MS analyses of RP–HPLC fractions followed by western blot analysis of the *N. nigricollis* venom fractions against the Pan-African EchiTAb-Plus-ICP antivenom (raised against venoms of *Echis ocellatus*, *Bitis arietans*, and *N. nigricollis*). Results indicated that at higher antivenom dilutions (1:4000 and 1:1000), the RP–HPLC fractions containing CTXs were not recognized by the antivenom. In contrast, at the same dilutions, there was significant recognition of the RP–HPLC fractions containing the higher molecular mass proteins of *N. nigricollis* venom. Nevertheless, at lower dilutions (1:500), the antivenom could react with the CTXs [25], which suggests that a higher dose of antivenom may be required to neutralize the toxic effects of the CTXs.

In another interesting study, Laustsen et al. [132], by ELISA, assessed the immuno-recognition of RP–HPLC fractions of *N. kaouthia* venom against immunoglobulins isolated from a human donor who had repeatedly immunized himself with several snake venoms (including those from three *Naja* species—*N. naja*, *N. kaouthia*, and *N. siamensis*). As expected, the human IgGs exhibited relatively poor immuno-recognition of the *N. kaouthia* venom fractions predominated by CTXs compared to those with higher molecular mass toxins [132], thereby highlighting the poor antigenicity of low-molecular mass toxins.

Immunological profiling of *Naja* venoms using the antivenomics approach has also revealed important information on recognizing cobra venom CTXs by homologous and heterologous antivenoms. In particular, CTXs from African spitting cobra (*N. nigricollis*, *N. katiensis*, *N. nubiae*, *N. mossambica*, and *N. pallida*) venoms exhibited variable recognition against the Pan-African EchiTAb-Plus-ICP antivenom (which contains *N. nigricollis* venom in the immunization mix). Moreover, the immuno-recognition of the CTXs was found to be isoform-specific, i.e., while a few isoforms were effectively immuno-captured by the antivenoms, some were partially recognized (10 to 80% immuno-capture), and some others were not recognized at all. Notably, the antivenom (at an eight-fold molar excess ratio) could not immuno-recognize several CTX isoforms of even the homologous *N. nigricollis* venom as revealed by the antivenomics analysis [25].

In the case of the heterologous venoms, the immuno-recognition was as poor as expected; several RP–-HPLC fractions of the venoms containing CTXs could not be recognized by the EchiTAb-Plus-ICP antivenom [25]. Similarly, Xu et al. [36] reported partial immuno-capture of *N. kaouthia* CTXs by the heterologous monovalent *N. atra* antivenom manufactured by Shanghai Serum Biological Technology Co. Ltd., China. Affinity chromatography-based immuno-profiling of the *N. naja* venoms from western and eastern India revealed that the majority of the 3FTXs, including CTXs, were not efficiently immuno-recognized by the homologous Indian polyvalent antivenom (raised against venoms of *N. naja*, *D. russelii*, *Echis carinatus*, and *Bungarus caeruleus*) [22,23]. A similar finding was also reported for the *N. kaouthia* venom from eastern India against the heterologous Indian polyvalent antivenom [23]. To sum up, immunological profiling highlights the poor immuno-recognition of the cobra venom CTXs, which is a severe concern for the efficient in-patient hospital management of cobra-envenomed patients.

While testing the immuno-recognition is undoubtedly an efficient way of assessing antivenom efficacy against venom toxins, neutralization of toxin function, either in vitro or in vivo, is also very important to understand the outcomes of antivenom treatment during the snake envenomation. In a recent study, Chong et al. [133] purified CTXs from three species of Asiatic cobras—*N. kaouthia* from Thailand, *N. sumatrana* from Malaysia, and *N. atra* from Taiwan—and investigated the cytotoxicity neutralization of these toxins towards CRL-2648 mouse fibroblast cell lines by the Vietnamese *N. kaouthia* monovalent antivenom (SAV), Thai *N. kaouthia* monovalent antivenom (NkMAV), and Taiwanese neuro bivalent antivenom (NBAV raised against Taiwanese *B. multicintus* and *N. atra* venoms). The results indicated that all three antivenoms (125 µg/mL) could neutralize the cytotoxicity of the tested cobra venom CTXs, albeit with variable potencies. While SAV at a dose of 125 µg/mL could neutralize the cytotoxicity of *N. kaouthia* CTXs (10 to 100 µg/mL) towards CRL-2648 mouse fibroblasts, the CTXs from *N. sumatrana* and *N. atra* required an increased quantity of the antivenom (500 µg/mL) for their effective neutralization. The other two antivenoms, NkMAV and NBAV, could also neutralize the CTXs of the cobra venoms, but only at a higher dose of 500 µg/mL. In terms of neutralization potency (mg of CTX neutralized/g of antivenom), all three antivenoms exhibited comparable potencies (~23 to 50 mg/g) against *N. sumatrana* and *N. atra* CTXs. However, the venom of the former species is not included in the immunization mixture of any tested antivenoms. Moreover, for neutralization of *N. kaouthia* CTXs, the potency of homologous NkMAV (13.2 mg/g) and heterologous NBAV (4.5 mg/g) were sub-optimal as compared to SAV (42.6 mg/g) [133]. These findings underscore the variable inter-species neutralization of CTXs by homologous and heterologous monovalent or bivalent antivenoms, with some antivenoms requiring increasing doses for efficient neutralization of cytotoxicity. The variability in neutralization potency might be due to variable CTX abundances in the venoms, or the formulation of the tested antivenoms (monovalent or bivalent). More importantly, increasing doses of antivenom for effective cobra venom CTX neutralization translates into a requirement for more antivenom vials in cobra-envenomed patients, which inevitably increases the risk of developing adverse serum reactions in the treated patients [134].

CTXs usually have a higher LD_50_ value (1.5 to 2.0 µg/g), albeit they exhibit cytolysis of mammalian cultured cells under in vitro and intense necrosis (or dermonecrosis) under in vivo conditions. However, the efficacy of antivenom in neutralizing the local effects of CTXs in cobra-envenomed patients has not been very well established. Nevertheless, a few studies have tried to understand the neutralization of dermonecrosis by commercial antivenoms in animal models.

Interestingly, findings from these studies suggest variable neutralization of lethality and dermonecrosis inflicted by cobra venom CTXs. For instance, Tan et al. [135] reported effective lethality neutralization of purified Thai *N. kaouthia* CTXs by the Thai *Naja kaouthia* monovalent antivenom (NkMAV) (normalized potency of 20.44 mg CTX/g of antivenom). However, they demonstrated poor neutralization (normalized potency of 7.37 mg CTX/g of antivenom) of the CTXs by the heterologous Australian CSL sea snake antivenom under in vivo conditions. Similarly, the EchiTAb-Plus-ICP antivenom was shown to effectively neutralize the dermonecrosis exhibited by the homologous *N. nigricollis* venom (60 µL antivenom/1 Minimum Necrotic Dose, MND) and heterologous *N. katiensis* (60 µL antivenom/1 MND) and *N. mossambica* (65 µL antivenom/1 MND) venoms. However, in vitro immunological profiling suggested partial immuno-recognition of CTXs of these African spitting cobras [25]. Since both in vitro and in vivo experiments employed a similar venom:antivenom ratio (~1:85), a plausible explanation for this discrepancy is that perhaps the major CTX isoforms of the abovementioned African cobra venoms that are responsible for inducing necrosis are indeed recognized and thereby neutralized by the EchiTAb-Plus-ICP antivenom. This signifies that although in vitro tests provide a useful way of determining the neutralizing potency of antivenoms against CTXs, the neutralization of dermonecrosis should also be tested in suitable animal models.

Surprisingly, the scenario is entirely the opposite in the case of the Chinese cobra *N. atra*. The freeze-dried neurotoxic antivenom (FNAV) (raised against *B. multicinctus* and *N. atra*) could not protect littermate ICR (CD1) mice against dermonecrosis induced by the crude *N. atra* venom or its purified CTXs. However, it efficiently neutralized the venom-induced lethality in the CD-1 mice (neutralization potency of 47.8 mg/mL of antivenom) [96]. These data clearly indicate that *N. atra* CTXs, due to a poor immunogen could not be neutralized by commercial antivenom. The discrepancy in neutralization of Asian and African cobra venom CTX-induced dermonecrosis may be due to the differences in composition and potency of the CTXs present in these cobra venoms. In this regard, it has been found that the necrotic activity of venoms of spitting cobras (mainly from Africa) is more pronounced compared to the non-spitting species (mostly from Asia), perhaps due to the fact that African spitting cobras contain a higher proportion of PLA_2_, particularly basic PLA_2_ in their venom than the non-spitting Asian cobras [136].

Another critical aspect of these neutralization studies of local effects induced by cobra venoms/CTXs is the choice of the experimental model. While all of the rescue models as mentioned in the above experiments used a pre-incubated venom/CTX-antivenom mixture for administration into the animals for testing the neutralization of dermonecrosis, the actual scenario of envenomation is quite distinct where antivenom is administered only after envenomation. In a recent study, Dutta et al. [72] demonstrated that the extent of neutralization of cytotoxicity of a PLA_2_–CTX–NTX cognate complex by Indian polyvalent antivenom to rat L6 myogenic cells is significantly lower when the antivenom is added exogenously 60 to 240 min post-treatment of the complex compared to pre-incubation of the complex with antivenom. Rivel and colleagues [102] also reported a similar observation for neutralizing purified CTXs isolated from *N. nigricollis* venom. These findings indicate that once the CTXs interact and bind to their pharmacological targets, it becomes relatively complex for the antivenoms to reverse the deleterious effects of the CTXs. This scenario aptly mimics envenomation in victims. On the contrary, if the CTXs are pre-incubated with antivenom in an isolated system, the antibodies present in the antivenom can recognize and thereby neutralize the toxins to some extent. Therefore, apart from poor neutralization of CTXs due to their low immunogenicity, the bioavailability of these venom toxins (see below) for neutralization is also an important parameter that determines the extent of neutralization of the CTXs in envenomed patients.

Unfortunately, CTXs exhibit high binding affinity at the envenomed site (cell membranes), resulting in a lower systemic bioavailability of these toxins in blood to the administered antivenom [122,123]. Therefore, poor immunogenicity of the CTXs, along with their lower systemic bioavailability, eventually results in sub-optimal neutralization of CTXs in envenomed victims. 

In a recent pilot study by Lin et al. [137] involving four confirmed cases of *N. atra* snake envenomation, it was evident that elevated concentrations of CTXs could be detected in the wound discharge fluid, necrotic tissue, and bullae of the patients even after ten days of administration of high doses of specific antivenom (bivalent antivenom raised against *B. multicinctus* and *N. atra*). More importantly, although the patients recovered from neuroparalysis, there was no observable recovery of dermonecrosis at the proximities of the bite site. The wound gradually worsened even after the administration of additional vials of antivenom. This adverse clinical symptom prompted the attending physicians to perform surgical debridement and excision of the necrotic lesion (median of 4th day) to avert wound deterioration [137]. A few other studies have also highlighted the requirement of surgical intervention for removing necrotic lesions (caused mainly by CTX) in about 60% of cobra-envenomed patients [138,139,140]. Therefore, these findings indicate that although antivenoms can exhibit moderate to low recognition and neutralization of CTXs under in vitro or in vivo experimental settings, neutralizing these toxins in patients is still a grave concern for in-patient hospital management of cobra-envenomation.

## 7. Conclusions and Future Perspectives

CTXs are among the most abundant components of cobra venoms. In some African cobras, they comprise more than 70% of the venom proteome. Being less toxic than cobra venom NTXs, CTXs act non-specifically, thereby affecting various tissues and organs in the body of the envenomed victim. Apart from inducing local necrosis, the other main target of CTXs is the cardiovascular system, and therefore, they also function as CdTxs. The ability of CTXs to form cognate complexes with other venom components leads to the potentiation of the toxic effects of the individual toxins and increases the magnitude of the deleterious effects of cobra envenomation, thus suggesting the importance of designing novel antivenoms targeting the least neutralized toxic components of venom such as CTX. However, apart from a few studies, not much has been revealed for CTX complexes and their role in cobra venom-induced pathophysiology, which therefore urges further research into this intriguing domain of CTX research. CTXs are fairly low-molecular-mass proteins and induce low immune response during the traditional production of antivenoms, which so far are the most effective treatment for snake envenomation. Nevertheless, the recent advances in protein engineering can greatly facilitate the solution to this problem and aid in creating highly immunogenic toxins/toxin fragments for antivenom production. In addition, topical application of antivenom (small antibodies, e.g., VHH or nanobodies) or small molecule inhibitors may be more effective to mitigate the toxic effects of cobra venom CTXs retained at the bite site. On the other hand, CTXs possess some favorable characteristics as they manifest a higher cytotoxicity to cancer cells compared to normal cells. This property can be exploited in the design of new anticancer medicines; however, the difference in cytotoxicity between normal and cancer cells have to be increased, thus there is a need for further studies on the precise molecular mechanisms of CTX effects. Moreover, being one of the predominant classes of toxins in cobra venoms, CTX antibodies can also be developed as markers for identification of cobra-envenomation. So, while CTXs were discovered a fairly long time ago, they are not as well studied as other cobra venom toxins including neurotoxins and phospholipases A_2_. Further studies on the molecular mechanisms of CTXs may result in the discovery of novel means for the management of cobra-envenomed patients and the design of novel anticancer drugs based on these molecules.

## Figures and Tables

**Figure 1 toxins-14-00839-f001:**
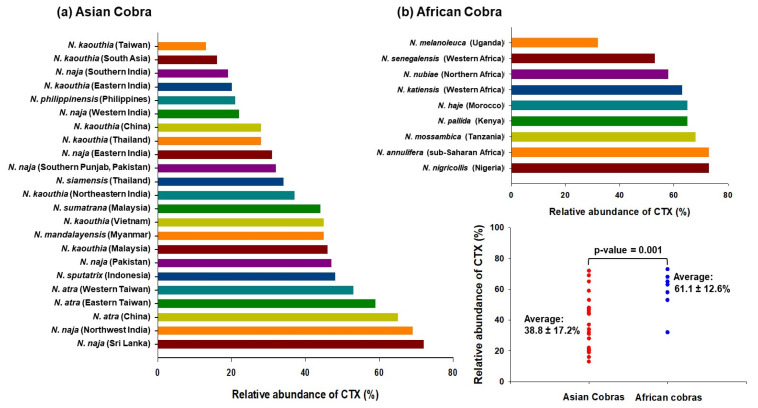
Proteomics analyses determined the relative abundance of CTXs in cobra venoms from diverse geographical locations [22,23,24,25,26,27,33,34,35,36,37,38,39,40,41,42,43,44,45,46,47]. The mean relative abundance of CTX in African cobra venoms (61.1 ± 12.6%) is significantly higher as compared to Asian cobra venoms (38.8 ± 17.2%); *p*-value = 0.001.

**Figure 2 toxins-14-00839-f002:**
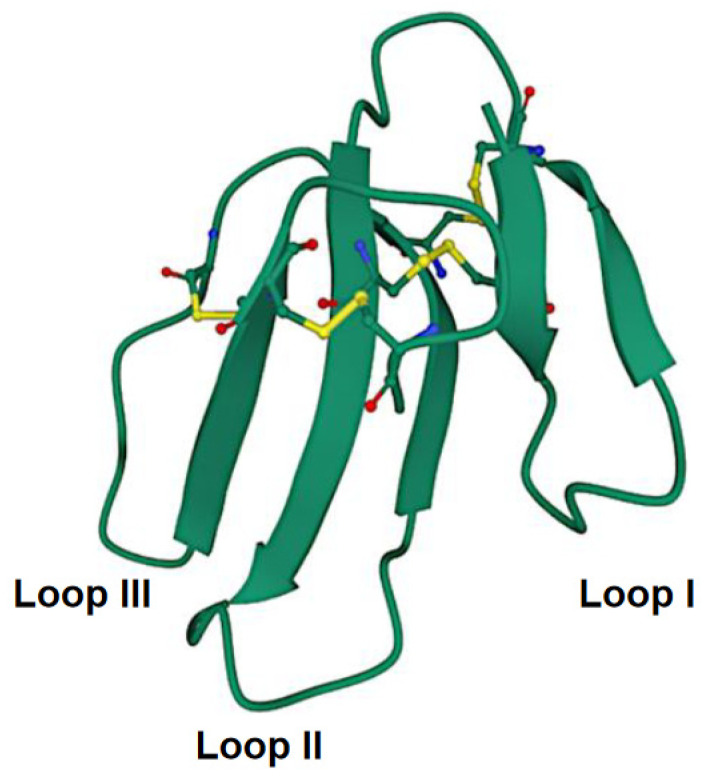
Crystal structure of cytotoxin 13 from *Naja naja*, hexagonal form (PDB 7QHI) (Dubovskii et al. [57]). Arrows indicate β-structures. Disulfide bonds are shown as balls and sticks. Accessed on 29 September 2022.

**Figure 3 toxins-14-00839-f003:**
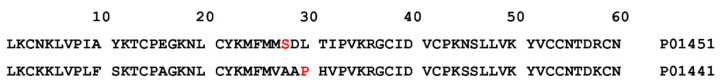
Amino acid sequences for CTXs of S-type and P-type. P01451 (SwissProt 3SA1_NAJOX): S- type cytotoxin 1 from *Naja oxiana*; P01441 (SwissProt 3SA2_NAJOX): P-type cytotoxin 2 from *Naja oxiana*. Serine-28 and proline-30 are shown in red.

**Figure 4 toxins-14-00839-f004:**
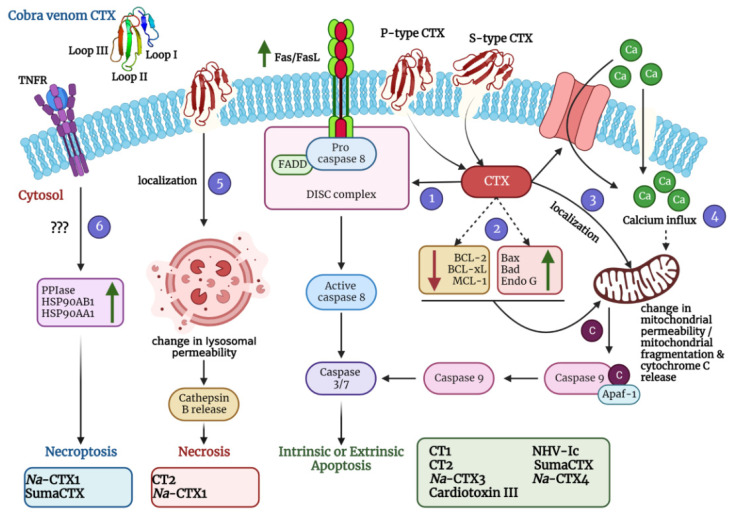
A schematic diagram depicting diverse mechanisms of action of cobra venom CTXs was created using BioRender.com. Molecular dynamics simulations with representative members of S-type CTX have indicated that this class of CTXs incorporate loop I or both loops I and II into the lipid bilayer membrane for cellular entry. On the contrary, P-type CTXs interact with the lipid membranes via all three loops (I-III). After pore formation and entry into the cells, CTXs can adopt various mechanisms that eventually lead to apoptosis, necrosis, and necroptosis of the affected cells. They can induce the upregulation of FasL/Fas expression (via activation of the Ca^2+^/NOX4/ROS/p38 MAPK signaling axis) thereby activating the extrinsic pathway of apoptosis (Pathway 1). They can also induce the upregulation of apoptotic proteins and a concomitant downregulation of anti-apoptotic proteins to stimulate intrinsic apoptosis (Pathway 2). Moreover, they can localize in the mitochondria and change the mitochondrial membrane permeability leading to release of cytochrome c and subsequent intrinsic apoptosis activation (Pathway 3). CTXs can also increase the influx of calcium ions either via calcium channels or through non-selective pores to perturb the mitochondrial membrane permeability thereby leading to intrinsic apoptosis (Pathway 4). They can also localize in the lysosomes and increase the release of cathepsin B for inducing necrosis (Pathway 5). In addition, CTXs can also activate the necroptotic cell death pathway via upregulation of peptidyl–prolyl isomerase (PPIase) and heat shock proteins (HSP90AB1 and HSP90AA1) (Pathway 6). For exact mechanism of individual CTX, please refer to Table 2. *Na*-CTX1, *Na*-CTX3, and *Na*-CTX4 represent CTX isoforms isolated from the venom of *N. atra*.

**Table 1 toxins-14-00839-t001:** Geographical distribution of different *Naja* species and their WHO medical importance category. Data presented in this table were retrieved from snakebite information and the data platform maintained by the World Health Organization (https://www.who.int/teams/control-of-neglected-tropical-diseases/snakebite-envenoming/snakebite-information-and-data-platform/overview#tab=tab_1). The data were accessed on 5 October 2022.

Snake Species	Common Name	Medical Importance/ Category	Region of Distribution	Number of Countries	Human Population (2020) in This Species’ Range
*N. anchietae*	Anchieta’s cobra	Highest, Secondary	Africa	6	19,008,230
*N. annulata*	Banded water cobra	Highest, Secondary	Africa	10	114,642,902
*N. annulifera*	Snouted cobra	Highest	Africa	8	70,731,878
*N. ashei*	Ashe’s spitting cobra	Highest	Africa	6	32,513,269
*N. atra*	Chinese cobra	Highest	Asia and Australasia	5	570,266,425
*N. christyi*	Christy’s water cobra	Secondary	Africa	3	15,111,896
*N. guineensis*	Black forest cobra	Highest, Secondary	Africa	7	55,106,930
*N. haje*	Egyptian cobra	Highest, Secondary	Africa	21	443,884,351
*N. kaouthia*	Monocled cobra, Thai cobra	Highest, Secondary	Asia	11	976,884,863
*N. katiensis*	Mali cobra, West Africa brown spitting cobra	Highest, Secondary	Africa	12	123,542,818
*N. mandalayensis*	Mandalay spitting cobra	Highest	Asia and Australasia	1	14,774,047
*N. melanoleuca*	Black and white cobra, Forest cobra	Highest, Secondary	Africa	11	244,375,176
*N. mossambica*	Mozambique spitting cobra	Highest, Secondary	Africa	10	130,049,980
*N. naja*	Indian cobra, Spectacled cobra	Highest, Secondary	Asia and Australasia	5	1,656,817,409
*N. nigricincta*	Western barred spitting cobra, Zebra cobra	Highest, Secondary	Africa	4	11,381,021
*N. nigricollis*	Black-necked spitting cobra	Highest, Secondary	Africa	33	727,256,279
*N. nivea*	Cape cobra	Highest	Africa	4	17,651,152
*N. nubiae*	Nubian spitting cobra	Secondary	Africa	5	39,843,095
*N. oxiana*	Central Asian cobra, Transcaspian cobra	Highest, Secondary	Asia and Australasia, Middle East	8	242,127,307
*N. pallida*	Red spitting cobra	Secondary	Africa	6	59,847,176
*N. peroescobari*	Sao Tome cobra	Highest	Africa	1	189,185
*N. philippinesis*	Northern Philippine cobra	Highest	Asia and Australasia	1	592,982,107
*N. sagittifera*	Andaman cobra	Secondary	Asia and Australasia	1	373,959
*N. samarensis*	Southern Philippine cobra, Visayan cobra	Highest	Asia and Australasia	1	30,350,207
*N. savannula*	West African banded cobra	Highest, Secondary	Africa	16	151,894,138
*N. senegalensis*	Senegalese cobra	Highest, Secondary	Africa	13	84,781,768
*N. siamensis*	Indochinese spitting cobra, Siamese spitting cobra	Highest, Secondary	Asia and Australasia	5	119,240,121
*N. sputatrix*	Southern Indonesian spitting cobra	Highest, Secondary	Asia and Australasia	2	167,089,984
*N. subfulva*	Brown forest cobra	Highest, Secondary	Africa	22	377,545,129
*N. sumatrana*	Equatorial spitting cobra	Highest, Secondary	Asia and Australasia	6	124,654,470

**Table 2 toxins-14-00839-t002:** List of CTXs purified from venoms of *Naja* species and their mechanism of action.

Cobra Species	Cytotoxin	UniProt ID	Mechanism	Methodology/Tested Model	References
*N. naja*	CTX1	P01447	Interaction with erythrocyte membrane via ‘head groove’ and ‘loop groove’ of loop II	Molecular dynamics simulation	[75]
	CT13Nn	A0A0U4N5W4	Transformation from the “water” conformation to the “membrane” conformation in loop II during insertion into lipid membranes	X-ray crystallography and molecular dynamics simulation	[57]
	CTX2a	P86538	Complex formation with PLA_2_ and NTX and entry into cells via specific binding of the PLA_2_ to Vimentin	L6 rat myogenic cells	[72]
*N. oxiana*	CT1	P01451	Insertion into lipid membranes primarily with either the tip of loop I or both ends of loops I and II	NMR spectroscopy and molecular dynamics simulation	[76]
			Contractions of papillary muscles	Cardiomyocytes from right ventricles of rat hearts	[77]
			Formation of non-selective pores in the cell membrane that facilitates the influx of Ca^2+^ and stimulation of cardiomyocyte contracture	Isolated rat heart	[66]
			Alteration of mitochondrial permeability and signaling, ultimately leading to the mitochondrial fragmentation and stimulation of intrinsic apoptosis	Bovine cardiomyocytes, MCF-7 breast cancer cells, Hep-G2 hepatocellular carcinoma cells	[78,79]
	CT2	P01441	Insertion into lipid membranes via immersion of loop I	Molecular dynamics simulation	[80]
			Contractions of papillary muscles	Cardiomyocytes from right ventricles of rat hearts	[77]
			Formation of non-selective pores in the cell membrane that facilitates the influx of Ca^2+^ and stimulation of cardiomyocyte contracture	Isolated rat heart	[66]
			Alteration of mitochondrial permeability and signaling, ultimately leading to mitochondrial fragmentation and stimulation of intrinsic apoptosis	Bovine cardiomyocytes, MCF-7 breast cancer cells	[78,81]
			Increase in lysosomal membrane permeability and cathepsin B protease activity, and necrosis	MCF-7 breast cancer cells, HepG2 liver cancer cells, DU-145 prostate cancer cells, HL-60 leukemia cells, MDCK Madin–Darby canine kidney cells	[79]
*N. atra*	Cardiotoxin 1/CTX1	P60304	Upregulation of FasL and Fas expression leading to extrinsic apoptosis	HL-60 and U937 leukemia cells	[82]
			Increase in lysosomal membrane permeability and cathepsin B protease activity, and necrosis	16HBE human bronchial epithelial cells, MCF-7 breast cancer cells, K562 and P388 leukemia cells, H22 liver cancer cells	[83]
			Increase in lysosomal membrane permeability and release of cathepsin B, and necroptosis	KG1a and HL-60 leukemia cells	[84]
	CTX A2	P01442	Interaction with low sulfated heparin domains of cell membrane for internalization	H9C2 rat cardiomyocytes and Chinese hamster ovary (CHO) cells	[85]
	CTX A4/CTX4	P01443	Interaction with fully sulfated heparin domains of cell membrane for internalization	H9C2 rat cardiomyocytes and Chinese hamster ovary (CHO) cells	[85]
			Activation of L-type calcium channels for the influx of Ca^2+^ and subsequent activation of calcium-dependent cardiomyocyte contraction	Rat aortic ring preparation	[86]
			ROS generation followed by alteration in mitochondrial permeability, cytochrome c release and activation of intrinsic apoptosis	SK-N-SH human neuroblastoma cells	[87]
	Cardiotoxin III/CTX3	P60301	Cell cycle arrest at sub-G1 stage	HL-60 leukemia cells	[88]
			Downregulation of cyclin B1, cyclin A, Cdc25C, and Cdk 1 expression	K562 leukemia cells and Ca9–22, SAS, and CAL27 oral squamous carcinoma cells	[89,90]
			Upregulation of pro-apoptotic proteins (Bad, Bax, endonuclease G) and downregulation of anti-apoptotic proteins (Mcl-1, Bcl-2, survivin, Bcl-XL and XIAP) leading to intrinsic apoptosis	Ca9–22 oral squamous cell carcinoma cells, MDA-MB-231 breast cancer cells, A549 lung cancer cells, colo 205 colorectal cancer cells, and K562 leukemia cells	[90,91,92,93,94]
			ROS generation followed by alteration in mitochondrial permeability, cytochrome c release and activation of intrinsic apoptosis	SK-N-SH human neuroblastoma cells	[87]
			Ca^2+^ influx, phosphorylation of AMPK, mitochondrial fragmentation, cytochrome c release, and intrinsic apoptosis	U937 leukemia cells	[95]
	RP-HPLC fraction containing CTX isoforms	Unavailable	Dermonecrosis	Littermate ICR (CD-1) mice	[96]
*N. kaouthia*	NK-CT1	P0CH80	Interaction with oligonucleotide–human DNA topoisomerase II alpha complex for arresting cell growth	Molecular modelling and docking	[97]
			Cell cycle arrest at sub-G1 stage	U937 and K562 leukemia cells	[98]
*N. haje*	NHV-Ic	P01389	Alteration of mitochondrial permeability and signaling, ultimately leading to mitochondrial fragmentation and stimulation of intrinsic apoptosis	1301 leukemia cells	[99]
*N. sumatrana*	SumaCTX	A0A7T7DMY7	Alteration of mitochondrial permeability and signaling, ultimately leading to mitochondrial fragmentation and stimulation of intrinsic apoptosis	MCF-7 breast cancer cells	[100]
			Upregulation of peptidyl–prolyl isomerase and heat shock proteins thereby leading to necroptosis	MCF-7 breast cancer cells	[101]
*N. nigricollis*	RP-HPLC fraction containing CTX and PLA_2_	Unavailable	Dermonecrosis	CD-1 mice	[102]

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
