# Peer review of "Current Insights in the Mechanisms of Cobra Venom Cytotoxins and Their Complexes in Inducing Toxicity: Implications in Antivenom Therapy"

_toxins, 2022, doi:10.3390/toxins14120839_

Round 1
Reviewer 1 Report
This manuscript presents an overview of the mechanisms of action and cytotoxic potential of Cobra venom toxins and their complexes derived from various species of Cobra around the world, as well as the potency of commercial antivenoms available on the market. There is a good flow of information in this manuscript, which is well-written. The author should consider the following suggestions:
1. Some typos need to be corrected, such as lines 17 and 356.
2. It would be useful to add a discussion of traditional remedies and peptide inhibitors for the treatment of envenomation to this review.
Author Response
This manuscript presents an overview of the mechanisms of action and cytotoxic potential of Cobra venom toxins and their complexes derived from various species of Cobra around the world, as well as the potency of commercial antivenoms available on the market. There is a good flow of information in this manuscript, which is well-written. The author should consider the following suggestions:
Reply: The authors thank the reviewer for his/her inspiring words and line by line reviewing of the manuscript. The manuscript is revised in the light of your suggestions.
Query/Suggestion 1: Some typos need to be corrected, such as lines 17 and 356.
Reply: The typos in lines 17 (Page 1; Abstract) and 356 (Page 6; Section 5.1.4; first paragraph) have been corrected in the revised manuscript.
Query/Suggestion 2: It would be useful to add a discussion of traditional remedies and peptide inhibitors for the treatment of envenomation to this review.
Reply: Thank you for your suggestion. However, we wish to inform the honourable reviewer that till date peptide inhibitors are primarily designed for enzymatic toxins mainly against phospholipase A2 enzymes and there is not much literature on peptide inhibitors for Cobra venom CTXs. We agree that a lot of work has been done on traditional remedies for treatment of snake envenomation. A recent review article has highlighted the traditionally used phytochemicals for the treatment of snake envenomation (10.1016/j.jep.2022.115208). However, this is a much focussed review on mechanistic insights of Cobra venom CTXs and inclusion of traditional remedies is beyond the scope of this review. You may kindly agree that inclusion of all the literature on traditional remedies would make the manuscript too much voluminous. We sincerely hope that our reasoning will be accepted by the honourable reviewer.
Reviewer 2 Report
1. A fairly complete and in-depth review article, covering the discussion of pharmacology/toxicology and immunology, should be a suitable article for reading and research reference.
2. In literature, it is a pity that this retrospective article lacks information about the biochemical properties of cytotoxins or studies on biochemical /genetic modifications to this protein.
Author Response
Query/Suggestion 1: A fairly complete and in-depth review article, covering the discussion of pharmacology/toxicology and immunology, should be a suitable article for reading and research reference.
Reply: Thank you for your encouraging words. We have tried our best to address your query pertaining to the review article.
Query/Suggestion 2: In literature, it is a pity that this retrospective article lacks information about the biochemical properties of cytotoxins or studies on biochemical /genetic modifications to this protein.
Reply: We wish to inform the honourable reviewer that unlike enzymatic venom components including phospholipase A2, proteases, and L-amino acid oxidases, there is a dearth of literature on biochemical studies on non-enzymatic Cobra venom CTXs. The basic biochemical properties including lack of enzymatic activity, overall charge of cytotoxins, and their diverse biological activities are already discussed in the Section 3. Further, although the role of certain amino acids in the structural stability of CTXs have been highlighted in a few studies by chemical modifications (already discussed in the review), reports on genetic modifications to explore the significance of the critical amino acids in CTXs are very limited. Nevertheless, we have briefly discussed the findings of a mutational analysis in our revised manuscript (Page 3; Section 3; third paragraph).
Reviewer 3 Report
Section 4: Although CTx clearly work in conjunction with PLA2s, it is not evident that this is always, or even often, as part of a complex
Lines 333-335: It's not exactly clear what you mean here.
Lines 651-663: No citation in this paragraph - is this still a discussion of ref 117 ? If so, re-cite this
Lines 755-759: African spitters have a greater content of PLA2, particularly basic PLA2, than the non-spitting Asian cobras. This is likely related to the first comment above.
Lines 783 - 788: Good point about retention of CTx at bite site, probably also a consequence of their highly basic charge and the generally acidic nature of ECM components. You could make the point here that topical application of AV [small antibodies, e.g. VHH or nanobodies] or small molecule inhibitors, if/when they become available, may be more fruitful.
Author Response
Query/Suggestion 1: Section 4: Although CTx clearly work in conjunction with PLA2s, it is not evident that this is always, or even often, as part of a complex.
Reply: Thank you for your suggestion. We agree with your viewpoint, and further research on snake venom complexes are necessary for a better understanding of toxin synergism among CTX and PLA2. Therefore, we are not implying that CTXs always interact with phospholipase A2 enzymes or other venom components to exhibit their biological activities. In section 4, we have tried to discuss whatever literature is available on complex formation by cobra venom CTXs, since complex formation in snake venoms have significant pathophysiological relevance.
Query/Suggestion 2: Lines 333-335: It's not exactly clear what you mean here.
Reply: The sentence has been rephrased to make the meaning clearer in the revised manuscript (Page 6; Section 5.1.3; first paragraph).
Query/Suggestion 3: Lines 651-663: No citation in this paragraph - is this still a discussion of ref 117? If so, re-cite this
Reply: The relevant reference is now cited in the revised manuscript (Page 12; Section 6; fourth paragraph).
Query/Suggestion 4: Lines 755-759: African spitters have a greater content of PLA2, particularly basic PLA2, than the non-spitting Asian cobras. This is likely related to the first comment above.
Reply: Thank you for this valuable suggestion. We have now included this in the revised manuscript (Page 13; Section 6; second paragraph).
Query/Suggestion 5: Lines 783 - 788: Good point about retention of CTx at bite site, probably also a consequence of their highly basic charge and the generally acidic nature of ECM components. You could make the point here that topical application of AV [small antibodies, e.g. VHH or nanobodies] or small molecule inhibitors, if/when they become available, may be more fruitful.
Reply: Thank you for your suggestion. We have now mentioned this point in the Conclusion and future perspectives section of the revised manuscript (Page 14; Section 7).
Reviewer 4 Report
In this manuscript, the authors (unknown) reviewed the mechanism of action of Cobra venom cytotoxins (CTXs) and their complexes, highlighting their significant role in the pathophysiology and toxicity of Cobra venom. First, they have discussed the different discovery stages of how and when the components of Cobra venom were identified, characterized, and studied in detail by different research groups. After that they have conversed about the complex formation of CTXs with other components of venoms and did a comprehensive discussion on how CTX and its complexes engage in different mode of action during venom-induced pathophysiology and toxicity. Further, they have discussed the potency of commercial antivenoms in reversing the adverse effects of Cobra venom cytotoxins and their complexes in envenomed victims.
Overall, the review is quite comprehensive, well designed and covered several significant aspects of the Cobra venom CTXs. It will further influence research community to discover a novel means for the management of the Cobra-envenomed patients, and the design of novel anticancer drugs based on these CTXs. So, I believe the manuscript has a potential to published in Toxins.
Comments
1) No author information is provided with the manuscript which is quite usual unless it is double blind review.
2) I would suggest authors should more frequently cross-refer their figures and tables in the main text. The cross-referring figures in text will help to visualize and comprehend text better, especially, the mechanism part.
Author Response
In this manuscript, the authors (unknown) reviewed the mechanism of action of Cobra venom cytotoxins (CTXs) and their complexes, highlighting their significant role in the pathophysiology and toxicity of Cobra venom. First, they have discussed the different discovery stages of how and when the components of Cobra venom were identified, characterized, and studied in detail by different research groups. After that they have conversed about the complex formation of CTXs with other components of venoms and did a comprehensive discussion on how CTX and its complexes engage in different mode of action during venom-induced pathophysiology and toxicity. Further, they have discussed the potency of commercial antivenoms in reversing the adverse effects of Cobra venom cytotoxins and their complexes in envenomed victims.
Overall, the review is quite comprehensive, well designed and covered several significant aspects of the Cobra venom CTXs. It will further influence research community to discover a novel means for the management of the Cobra-envenomed patients, and the design of novel anticancer drugs based on these CTXs. So, I believe the manuscript has a potential to published in Toxins.
Reply: Thank you for your encouraging words. We have tried our best to address your suggestions.
Comments
Query/Suggestion 1: No author information is provided with the manuscript which is quite usual unless it is double blind review.
Reply: We believe the journal follows a double blind review policy and therefore the author information is not provided.
Query/Suggestion 2: I would suggest authors should more frequently cross-refer their figures and tables in the main text. The cross-referring figures in text will help to visualize and comprehend text better, especially, the mechanism part.
Reply: Thank you for your valuable suggestion. We have now tried to frequently cite the figures and tables in the main text of the revised manuscript (Page 6; Section 5.1.2, Page 8; Section 5.3, Page 9; Section 5.3 & 5.4, Page 10; Section 5.4).